Review

Subject Area:
developmental biology

Keywords:
organ regenerative therapy, ectodermal organ, organ germ method, organoid

Author for correspondence:
Takashi Tsuji
e-mail: takashi.tsuji@riken.jp

# Functional ectodermal organ regeneration as the next generation of organ replacement therapy

Etsuko Ikeda[1], Miho Ogawa[1,2], Makoto Takeo[1] and Takashi Tsuji[1,2]

[1]Laboratory for Organ Regeneration, RIKEN Center for Biosystems Dynamics Research, Kobe, Hyogo 650-0047, Japan
[2]Organ Technologies Inc., Tokyo 101-0048, Japan

EI, 0000-0003-3903-8885; MO, 0000-0003-1160-7273; TT, 0000-0002-2230-8052

In this decade, substantial progress in the fields of developmental biology and stem cell biology has ushered in a new era for three-dimensional organ regenerative therapy. The emergence of novel three-dimensional cell manipulation technologies enables the effective mimicking of embryonic organ germ formation using the fate-determined organ-inductive potential of epithelial and mesenchymal stem cells. This advance shows great potential for the regeneration of functional organs with substitution of complete original function *in situ*. Organoids generated from multipotent stem cells or tissue stem cells via establishment of an organ-forming field can only partially recover original organ function owing to the size limitation; they are considered 'mini-organs'. Nevertheless, they hold great promise to realize regenerative medicine. In particular, regeneration of a functional salivary gland and an integumentary organ system by orthotopic and heterotopic implantation of organoids clearly points to the future direction of organ regeneration research. In this review, we describe multiple strategies and recent progress in regenerating functional three-dimensional organs, focusing on ectodermal organs, and discuss their potential and future directions to achieve organ replacement therapy as a next-generation regenerative medicine.

## 1. Introduction

Numerous advances in various research fields, including developmental biology, stem cell biology and tissue engineering technology, have facilitated regenerative medicine [1–3]. The first generation of regenerative medicine is stem cell transplantation therapy using tissue-derived stem cells, embryonic stem (ES) cells or induced pluripotent stem (iPS) cells [4–7]. For example, bone marrow transplantation has already become a common treatment for leukaemia and hypoplastic anaemia. In addition, both ES cells and iPS cells are entering clinical trials for many diseases and injuries, including leukaemia, Parkinson's disease and Alzheimer's disease, cardiac infarction, diabetes, liver disease and various other conditions [8–11]. Tissue regeneration is positioned as the second generation of regenerative medicine, and several products, including skin and cartilage, are already on the market. Furthermore, the world's first tissue regeneration therapy using iPS cells derived from either the patient or an anonymous donor is being investigated in a clinical trial to cure age-related macular degeneration [12,13].

The next generation of regenerative therapy targets entire organs composed of multiple cell types with a complex three-dimensional structure [14]. In this decade, advances in the field of stem cell biology and developmental biology have provided new opportunities to regenerate functional organs. During embryonic development, organs arise from the respective organ germs,

which are induced by reciprocal interactions between fate-determined epithelial and mesenchymal stem cells, according to individual organ-forming fields (figure 1a) [15]. Functional organ regeneration was first achieved in 2007 by developing a novel cell manipulation method to generate a bioengineered organ germ with organ-inductive potential epithelial and mesenchymal stem cells isolated from an embryonic organ germ (figure 1b) [16]. This pioneering study and subsequent studies reported the fully functional regeneration of multiple types of ectodermal organs, providing evidence for the concept of functional organ regeneration [17–21].

The next paradigm shift came in 2008 with the discovery of organoids, which were generated by inducing an organ-forming field in a cell aggregate arising from pluripotent stem cells such as ES cells and iPS cells, as well as tissue stem cells (figure 1c) [22]. Virtually all types of organoids can be generated, including those of the central nervous system (i.e. cerebral cortex, pituitary gland, optic cup and inner ear) [23–29]. Although the emergence of the organoid represents a technological breakthrough now serving as an essential tool in many basic biology and clinical applications, organoids still can only partially reproduce the structure and function of the original organs. Therefore, the majority of single organoids generated to date could substitute for limited and/or partial functions of a complete organ, and are thus currently considered as mini-organs. Recently, salivary gland organoids were successfully developed that demonstrate fully functional organ regeneration with orthotopic transplantation [30]. Because the principles of ectodermal organ development are similar to those of other organs, it is important to gain a deeper understanding of ectodermal organ regeneration to achieve the complete functional regeneration of other organs (figure 1a). Furthermore, regeneration of an integumentary organ system (IOS) using an *in vivo* organoid method clearly demonstrated the possibility for organ system regeneration [31].

In this review, we describe recent progress in organ regeneration using various stem cell populations and strategies based on developmental biology and stem cell biology and discuss the future directions for organ replacement therapy as the next generation of organ regenerative medicine.

## 2. Development of a three-dimensional cell manipulation method, the organ germ method, using embryonic cells

Researchers have attempted to regenerate organs for several decades by combining functional cells, scaffold materials and physiologically active substances using tissue engineering techniques [32,33]. Although these previous studies made certain contributions towards organ regeneration, considerable concerns exist regarding the findings from these studies, such as the low efficiency of organ induction and the uncontrollable direction and size of the regenerated organ. With advances in stem cell and developmental biology, the reproduction of organogenesis in the fetal stage has progressed over the past 30 years. The developmental process of organ regeneration starts with the induction of the organ germ by epithelial–mesenchymal interactions in the organ field that form after the establishment of the body plan during early development. Cell manipulation

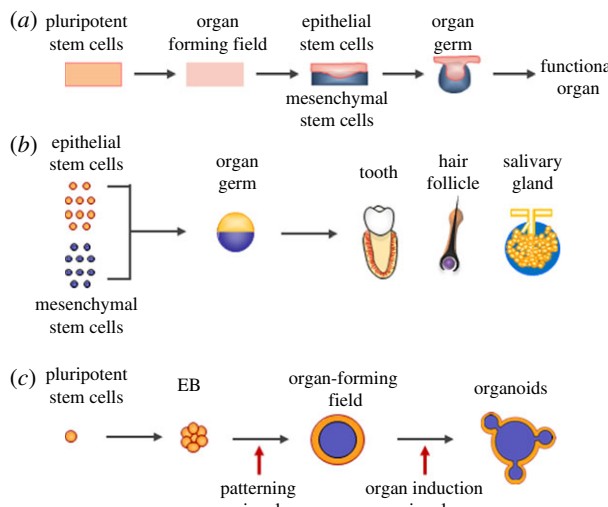

**Figure 1.** Schematic illustration of embryonic organogenesis and approaches for organ regeneration. (*a*) Schematic diagram of organogenesis. A functional organ is developed through the establishment of organ-forming fields, formation of organ germs by reciprocal epithelial and mesenchymal interactions, and morphogenesis. (*b*) Scheme of the fully functional regeneration of an ectodermal organ by mimicking organ germ formation using embryonic fate-determined epithelial and mesenchymal stem cells with organ-inductive potential. (*c*) Schematic illustration of organoid generation by recapitulating the establishment of organ-forming fields in cell masses generated from pluripotent stem cells.

techniques designed to regenerate organ germs have been developed over the years, but complete reproduction of the development and regeneration of functional organs has not been achieved [3,34].

We developed a bioengineering method, designated the organ germ method, to recapitulate the induction of the organ germ through epithelial and mesenchymal interactions in early developmental stages [16]. We compartmentalized epithelial and mesenchymal cells isolated from the mouse embryo at a high cell density in a type I collagen gel to achieve a precise replication of the processes occurring during organogenesis. Using this novel method, we have observed the functional regeneration of multiple types of ectodermal organs, such as teeth, hair follicles and secretory glands [17–21].

## 3. Fully functional bioengineered teeth

### 3.1. Tooth development

In tooth germ development, the dental lamina initially thickens (lamina stage) (figure 2a). The tooth germ develops and interacts with the oral mucosal epithelium and mesenchyme. Subsequently, epithelial thickening at the future location of the tooth and subsequent epithelial budding (bud stage) to the underlying neural crest-derived mesenchyme are induced by epithelial signals on embryonic days (EDs) 11–13 in mice. At EDs 13–15, the enamel knot acts as a signalling centre responsible for the formation and maintenance of the dental papilla. The primary enamel knots are formed at the tooth bud and appear during the transition from the bud to the cap stage. At EDs 17–19, the epithelial and mesenchymal cells in the tooth germ terminally differentiate [35–37]. The mesenchyme also differentiates into dental pulp and

royalsocietypublishing.org/journal/rsob    Open Biol. **9**: 190010

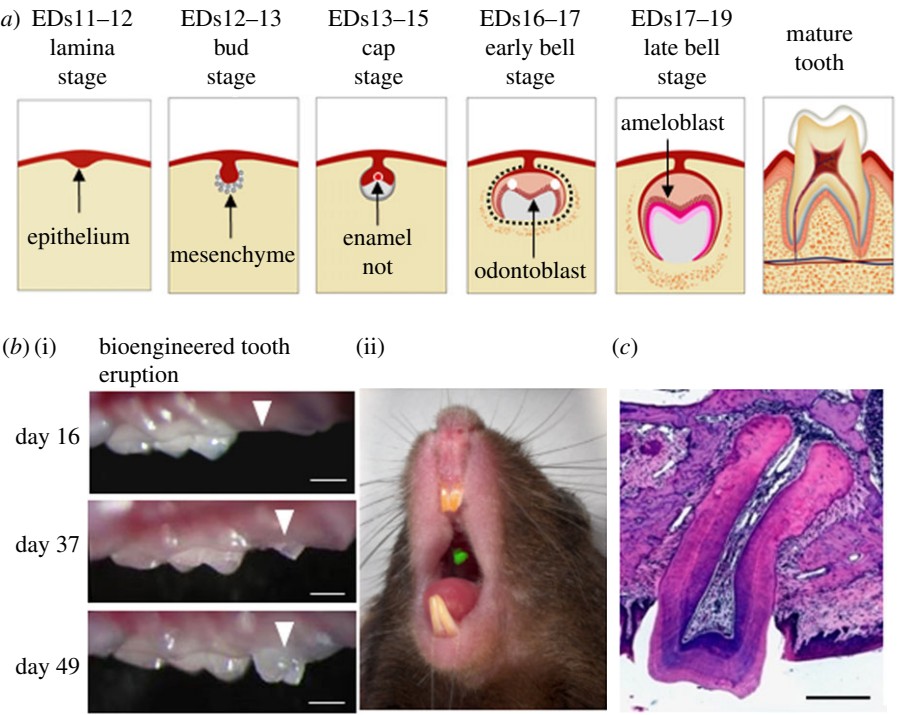

**Figure 2.** Fully functional bioengineered teeth regeneration. (*a*) Scheme of tooth development. (*b*) Time course analysis of tooth eruption from transplanted bioengineered tooth germ (i) and regenerated tooth using GFP-transgenic mouse-derived cells (ii). Scale bar: 500 μm. (*c*) Histological analysis of regenerated tooth. Note that the bioengineered tooth also formed a correct tooth structure, which comprised enamel, dentin, dental pulp and periodontal tissue. Scale bar: 200 μm.

periodontal tissues, which will become the cementum, periodontal ligament and alveolar bone. Tooth root formation is initiated after tooth crown formation, and the mature teeth erupt into the oral cavity.

## 3.2. Fully functional tooth regeneration

Tooth loss due to dental caries, periodontal disease or trauma causes fundamental problems with proper oral function and are associated with oral and general health issues [38]. Conventional dental treatments designed to restore occlusal functions after tooth loss are based on replacing teeth with artificial materials, such as fixed or removable dentures and bridge work. Although these artificial therapies are widely applied to treat dental disorders, the recovery of an occlusion is necessary because the teeth coordinate with the occlusal force and orthodontic force of the surrounding muscles, and integrity of the stomatognathic system is retained by establishing the occlusal system during jaw growth in the postnatal period [39–42]. Recent advances in tissue regeneration have enabled researchers to enhance the functions of biological teeth by facilitating underlying tooth development through bone remodelling and aiding the ability to perceive noxious stimuli [43].

As shown in our previous study, a bioengineered tooth germ develops into the correct tooth structure and successfully erupts into the oral cavity after transplantation into the region of the lost tooth (figure 2*b*) [17]. In the case of a transplanted bioengineered mature tooth unit comprising a mature tooth, periodontal ligament and alveolar bone can be engrafted into the tooth loss region through bone integration in the recipient (figure 2*c*) [18]. The bioengineered tooth maintains interactions with the periodontal ligament and alveolar bone originating from the bioengineered tooth unit through successful bone integration. The hardness of

the enamel and dentin of the bioengineered tooth components were within the normal range when analysed using the Knoop hardness test [17,18]. As a future direction, control of the tooth form is considered to be important. Teeth are generated by guiding the mesenchyme according to the body plan during the development process. Regarding tooth morphological control, the tooth width is controlled by the area of contacts between epithelial and mesenchymal cell layers, and the number of cusps is controlled by the expression of Shh in the inner enamel epithelium [44]. This bioengineered tooth technology contributes to the realization of whole-tooth replacement regenerative therapy as a next-generation therapy.

# 4. Fully functional bioengineered hair follicle

## 4.1. Hair follicle development

Mice have four different types of hair on their backs, classified as guard, awl, auchene and zigzag hairs. Hair follicle development in the mouse back skin begins with the fate determination of mesenchymal cells at approximately ED 10.5, resulting in the formation of a dermal condensate. Reciprocal interactions between the dermal condensate and overlying epidermis lead to the induction of the hair placode (figure 3*a*). Once the hair placode is established, hair follicle development occurs in three waves, starting with the development of the guard hair at ED 14.5, followed by awl and auchene hair at ED 17 and zigzag hair at birth [45,46]. The lower end of the hair peg epithelium that wraps around a condensed dermal cell forms the germ of the hair matrix. The condensed dermal cell forms a dermal papilla, which is considered a niche for hair follicle mesenchymal stem cells,

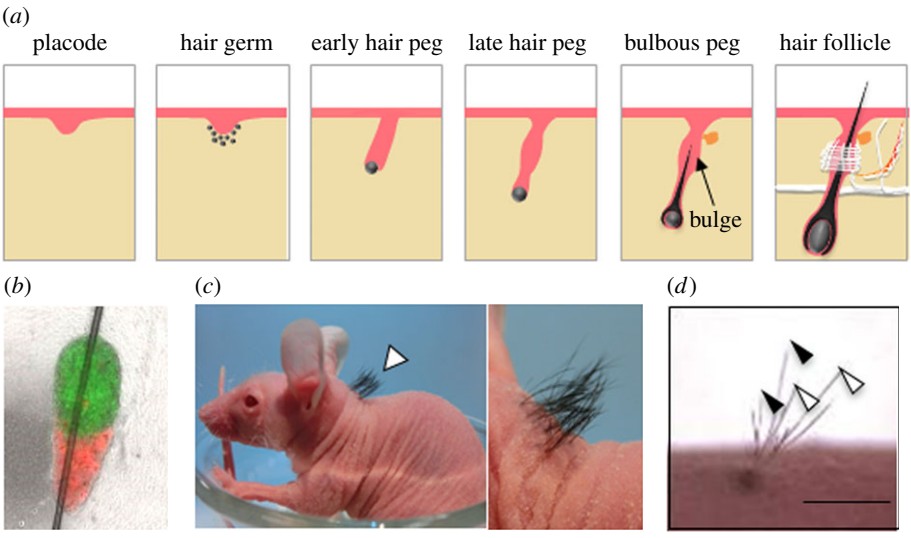

**Figure 3.** Fully functional bioengineered hair follicle regeneration. (*a*) Scheme of hair follicle development. (*b*) Representative bioengineered organ germ generated from bulge epithelial cells (green) and dermal papilla sells (red). (*c*) Macro-morphological observations of the bioengineered hairs (arrowhead). (*d*) Analysis of the piloerection capability by acetylcholine (ACh) administration. White arrowhead, before ACh injection; black arrowhead, after ACh injection. Scale bar: 1 mm.

and induces the differentiation of the hair matrix and forms the inner root sheath of the hair follicle and the hair shaft. The bulge region also forms an epithelium stem cell niche and simultaneously connect with nerve fibres and the arrector pili muscle (figure 3*a*) [47,48].

## 4.2. Fully functional hair follicle regeneration

The hair organ has the biological functions of thermoregulation, physical insulation from UV radiation, waterproofing, tactile sensation, protection from noxious stimuli, camouflage and social communication [49,50]. Hair loss disorders, such as congenital hair follicle dysplasia and androgenetic alopecia, are psychologically distressing and have negative effects on the quality of life of both sexes [51]. Current pharmacological treatments are insufficient in achieving ideal control of hair loss, such as congenital hair follicle dysplasia or alopecia areata [51]. The demand for the development of bioengineering technologies that enable regenerative therapy for hair loss has increased.

In the hair cycle, hair follicle germs are reconstituted periodically, and epithelial and mesenchymal stem cells capable of regenerating hair follicles are present, even in adults. Therefore, this organ is the only organ for which reconstituting germs can be regenerated from adult-derived cells. Autologous hair follicle transplantation in which a single hair follicle is isolated from the healthy scalp region and transplanted into patients with male pattern alopecia has been reported, and the transplanted hair follicles retain their characteristics [52]. According to many researchers, the replacement of dermal cells in skin using mesenchymal cells, which are collected from adult hair bulbs in a hair follicle, induces new hair follicle formation [53,54]. However, the regeneration of hair follicles that function in cooperation with the surrounding tissue is difficult. Our group reconstituted a bioengineered hair follicle germ, which contains mesenchymal stem cells, using bulge-derived epithelial cells and dermal papilla cells isolated not only from embryos, but also from adult mice

(figure 3*b*) [19]. After orthotopic transplantation, the bioengineered hair follicle germs develop into mature hair follicles with proper structures and produce hair throughout their life (figure 3*c*). Moreover, the regenerated hair follicles connected efficiently with the surrounding host tissue and showed pilomotor reflex in response to acetylcholine administration (figure 3*d*). This study demonstrated the potential of tissue stem cells isolated from adult hair follicles to develop into human hair follicles in the field of regenerative medicine.

# 5. Fully functional bioengineered secretory glands

## 5.1. Salivary and lacrimal gland development

Secretory glands, including salivary glands and lacrimal glands, are vital for the protection and the maintenance of physiological functions in the microenvironment of the oral and ocular surfaces. Secretory glands develop via reciprocal epithelial–mesenchymal interactions [55,56]. Salivary glands are classified into three major types: the parotid gland (PG), submandibular gland (SMG) and sublingual gland (SLG). The SMG develops through the invagination of the epithelium into the mesenchymal region on ED 11. The invaginated epithelial tissue proliferates to form an epithelial stalk (figure 4*a*) [57,58]. A terminal bud forms a branched structure by developing a cleft and by repeating the elongation and branching process from EDs 12.5–14.5 [59–61]. The terminal bulbs differentiate into acinar cells and mature to synthesize secretary proteins on ED 15 [62]. By contrast, the lacrimal gland also develops through the invagination of the epithelium into a mesenchymal sac at a temporal region of the eye on ED 12.5. The rounded epithelial buds condense into the superior conjunctival fornix, which then invaginate into the surrounding mesenchyme [63]. The lacrimal gland germ forms branches via stalk elongation and cleft formation morphogenesis. The fundamental structure of the lacrimal gland is achieved by ED 19 [64].

royalsocietypublishing.org/journal/rsob    *Open Biol.* **9**: 190010

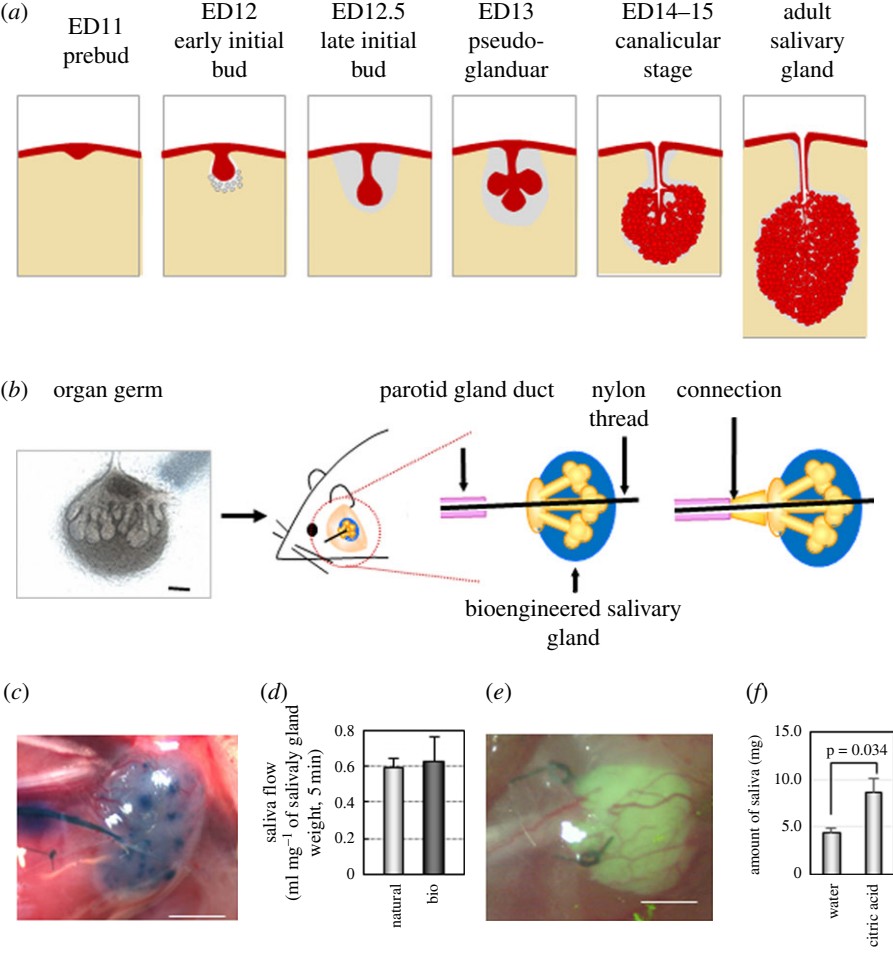

**Figure 4.** Fully functional bioengineered salivary gland regeneration from bioengineered organ germ and organoid. (*a*) Schematic of salivary gland development. (*b*) Schematic of the transplantation of the bioengineered SMG. The bioengineered germ was transplanted with a nylon thread to promote the connection to the duct at the location of the PG. Scale bar: 200 μm. (*c*) Photograph of a bioengineered SMG on day 30 after transplantation in a mouse with salivary gland defects. (*d*) Assessment of the amount of saliva secreted by normal mice (light bars) and bioengineered SMG-engrafted mice (dark bars) after gustatory stimulation with citrate. The data are presented as means ± s.e.m. Scale bar: 200 μm. (*e*) Photograph of GFP-labelled salivary gland-derived from mouse ES cells on day 30 after transplantation in a mouse with salivary gland defects. Scale bar: 200 μm. (*f*) Assessment of the amount of saliva secreted in combined salivary gland organoid-engrafted mice after stimulation by water (right bars) or citric acid (left bars). The data are presented as the means ± s.e.m.

## 5.2. Salivary and lachrymal gland regeneration

The dry mouth and dry eye are common symptoms. Salivary gland diseases include salivary tumours, obstructive disorders, infections and the symptoms of systemic diseases, such as Sjögren's syndrome, lymphoma and metabolic diseases [65–67]. These conditions also affect the lacrimal glands, resulting in dry eye [68–71]. Dysfunction and disorders associated with these exocrine glands result in a general reduction in the quality of life. However, current therapies for diseases characterized by dry mouth and eye only treat the symptoms [72,73]. These therapies only exert temporary effects and do not reverse exocrine gland dysfunction.

Our group aimed to develop better treatments by reconstituting a bioengineered salivary gland germ from epithelial and mesenchymal cells derived from ED 13.5–14.5 mouse embryonic salivary gland germs using our developed organ germ method (figure 4*b*) [20]. After orthotopic transplantation following the removal of native salivary glands, the bioengineered salivary gland germ developed into a mature salivary gland, and a proper connection was formed between the host salivary duct and the bioengineered salivary duct (figure 4*b*). This procedure led to the development of a connected salivary gland duct in the recipient mouse with acinar tissue structures that were similar to the natural salivary gland (figure 4*c*). The bioengineered SMG regenerated serous acinar cells and exhibited a natural organ structure. Nerve entry into these bioengineered salivary glands was also noted, and saliva secretion was induced in the salivary gland by taste bud stimulation using citric acid (figure 4*d*).

We also reconstituted a bioengineered lacrimal gland germ from epithelial and mesenchymal cells derived from the lacrimal gland germs of ED 16.5 mouse embryo [21]. The bioengineered lacrimal gland germ, which was generated using the organ germ method, successfully underwent branching morphogenesis. After transplantation, these glands developed into mature secretory gland structures *in vivo*. These results confirmed the possibility of regenerating a bioengineered secretory gland using organ germ transplantation.

## 6. Generation of organoids as mini-organs from pluripotent stem cells

Organoids, which reproduce the partial structure and function of organs, were generated from multipotent stem cells

based on the concept of recapitulating the induction process of an organ-forming field with subsequent self-organization during embryonic organogenesis. This induction was achieved by using various combinations of cytokines, which mimic the patterning and positioning of signalling in the embryo. This concept was first proven by the successful generation of an optic cup organoid from ES cells [22]. Subsequently, various organoids were induced in each organ-forming field, such as the retina [24], pituitary gland [26], cerebrum [27,28], inner ear [29] and hair follicle [74] in the head field; thyroid [75] and lung [76,77] in the thorax field; and small intestine [78], stomach [79] and kidney [80,81] in the abdomen field [82].

Adult tissue stem cells, such as intestinal [83], pulmonary [84], gastric [85,86] and pancreatic stem cells [87], are also capable of generating organoids through self-organization of their niche, which can partially reproduce the original tissue structure. Although the definition of an organoid is slightly different depending on its origin (i.e. pluripotent stem cells or tissue stem cells), organoids partially recapitulate the organ or tissue structure and can grow to a limited small size, and are thus considered mini-organs. Therefore, unlike a bioengineered organ germ, an organoid is incapable of completely substituting the functions of its original organs following orthotopic transplantation on its own; however, orthotopic and heterotopic transplantation of multiple organoids can partially recover organ function [6,77,88,89].

Recently, we successfully regenerated a fully functional salivary gland from mouse ES cells *in vivo* (figure 4*e,f*) [30]. Using the general method for organoid formation, we generated the salivary gland primordium as an organoid through the induction of an organ-forming field (i.e. the oral ectoderm), which was then transplanted orthotopically. The transplanted organoid developed into a mature salivary gland with the correct tissue structure such as acinar tissue, and formed appropriate connections with surrounding tissues, including the PG duct and nerves. Moreover, the regenerated salivary gland secreted saliva in response to taste stimulation using citric acid, demonstrating the full functional recovery of the original salivary gland following orthotopic transplantation of the organoid (figure 4*f*). These studies clearly demonstrate the feasibility of functional organ regeneration using organoids, generated by inducing organ-forming field in multipotent stem cell, not embryonic organ-inductive potential stem cells. Development of the novel *in vitro* culture system enabling organoids of large organs, such as liver and kidney, to grow to an appropriate size should be the next topic of research to achieve organ regeneration.

## 7. Regeneration of a three-dimensional IOS from iPS cells

The coordinated function of multiple organs, collectively referred to as an organ system such as the central nervous system, circulatory system, digestive system and IOS, is vital to sustaining homeostasis in an organism [90]. Therefore, regeneration of the entire organ system is the next challenge in the field of regenerative medicine. The IOS is the largest organ system in the body. This system contains several organs, such as hair follicles, sebaceous gland and sweat gland, in addition to the skin tissue that is composed of the epidermis, dermis and subcutaneous fat. The skin organ system plays important roles in homeostasis, such as secretion of moisture and sebum, and protection from ultraviolet light and external stimulation by hair shafts. Skin injury by severe burns is life threatening. Congenital defects and loss of skin appendages significantly affect the quality of life, although partial regenerative medical treatment with epidermal sheets is possible. The creation of artificial skin that comprises the epidermis and dermis, and regeneration of hair follicle organs through cell manipulation, have been reported. Nevertheless, no skin organ system has been regenerated.

Recently, we successfully regenerated the IOS by inducing an organ-forming field in embryoid bodies (EBs) derived from mouse iPS cells (figure 5*a*) [31]. After transplantation of EBs into the subrenal capsule, generation of skin appendages including hair follicles, sebaceous glands and subcutaneous adipose tissue was confirmed in the bioengineered IOS with no tumorigenesis (figure 5*b,c*). Furthermore, the number and density of regenerated hairs in the bioengineered IOS were the same as those found in natural hair, suggesting that organogenesis in the IOS occurred in a similar manner as in normal development. The bioengineered IOS generated in the subrenal capsule was fully functional after transplantation to the back skin of nude mice, as evidenced by the repetitive hair cycle (figure 5*d*). This study proved the concept of organ system regeneration *in vivo*. From a practical application perspective, a novel strategy to generate an organ system *in vitro* is desired. One such strategy could be to assemble the multiple types of organoids as parts. Research to control the configuration of organoids and grow them *in vitro* will be the next trend in the field of regenerative medicine.

## 8. Conclusion and future perspectives

In this decade, studies of organ regeneration starting from bioengineering technology have made large strides towards the realization of organ regenerative therapy by incorporating the concepts from stem cell biology and developmental biology. Based on the findings from organoid studies, virtually all mini-organs can be generated from either pluripotent stem cells or tissue stem cells, dispelling the concerns about the cell source for organ regenerative therapy. Functional regeneration of ectodermal organs using cells isolated from embryonic organ germ, organ-inductive potential stem cells and pluripotent stem cells prove the concept of organ replacement therapy.

The development of an *in vitro* three-dimensional culture system with the ability to grow organoids and organ germs to an appropriate size is essential to achieve the functional regeneration of multiple organs and organ systems. Current *in vitro* culture systems do not enable the appropriate growth or maintenance of organoids or organ germs due to the appearance of necrosis inside these tissues, mainly due to the lack of a nutrient supply. *In vivo*, the blood circulation system is essential to maintain organ functions through oxygen transport, nutrient supply and waste removal. Recent progress in tissue engineering has shown that the vascular network administers biological substances to the interior of the cell spheroid [91]. Moreover, we previously developed an organ perfusion culture system using a vascular network that maintained the rat liver in a healthy

royalsocietypublishing.org/journal/rsob Open Biol. 9: 190010

royalsocietypublishing.org/journal/rsob    Open Biol. 9: 190010

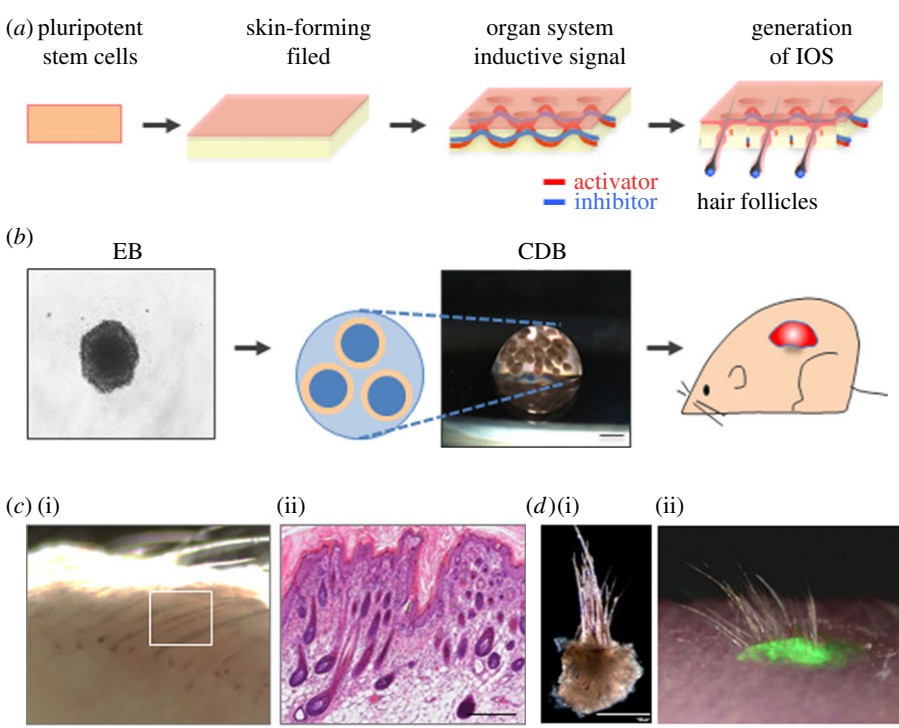

**Figure 5.** Bioengineering a three-dimensional IOS from iPS cells. (*a*) Scheme of IOS formation from pluripotent stem cells by inducing skin-forming fields and subsequent organ system inductive signals. (*b*) Schematic of EBs cultures and the novel transplantation method, a clustering-dependent EB (CDB) transplantation, in which EBs are spatially arranged into collagen gels to induce epithelial tissues. Scale bar: 50 μm. (*c*) Dissection microscopy (i) and H&E staining (ii) of iPS cell-derived bioengineered three-dimensional IOS. Scale bar: 500 μm. (*d*) Dissection microscopy of a skin fragment of the bioengineered IOS before (i) and after (ii) transplantation. Note that the eruption and growth of the hair shaft occurred after the transplantation of the skin fragment. Scale bar: 200 μm.

condition for an extended period [92], providing clues for the development of a novel three-dimensional culture system.

Because hair follicle stem cells are the only adult stem cells possessing organ-inductive potential that can be transplanted autogenously, the first human clinical trial of organ regenerative therapy will undoubtedly investigate hair follicle regeneration. The regeneration of hair follicles using our organ germ method is now being investigated in a pre-clinical study to cure patients suffering from androgenic alopecia, with an aim of conducting clinical trials in 2020. This hair follicle regenerative therapy will be a milestone in organ regenerative therapies and will lead to the development of material and responsive infrastructure to realize organ regenerative medicine. Applying knowledge of hair follicle regeneration and expertise obtained from clinical trials to other organ germs or organoids will enable the

regeneration of other organs from pluripotent and tissue stem cells in combination with organoid technologies in the next few decades.

Data accessibility. This article has no additional data.

Authors' contributions. T.T. designed this review. E.I., M.O., M.T. and T.T. wrote the manuscript.

Competing interests. This study was performed under an invention agreement between the Riken and Organ Technologies Inc. T.T. is a director at Organ Technologies Inc.

Funding. The publication of this review was partially supported by a Grant-in-Aid for KIBAN (A) from the Ministry of Education, Culture, Sports, Science, and Technology (grant no. 25242041) and by a collaboration grant (to T.T.) from Organ Technologies Inc. This work was partially funded by Organ Technologies Inc.

Acknowledgements. The authors thank the members of their laboratories who performed the experiments referenced in the manuscript.

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
