## [Reviewer comments · Open Biology]

Review History

RSOB-19-0010.R0 (Original submission)

Review form: Reviewer 1

Recommendation

Accept with minor revision (please list in comments)

Are each of the following suitable for general readers?

- a) **Title**
Yes
- b) **Summary**
Yes
- c) **Introduction**
Yes

Is the length of the paper justified?

Yes

Should the paper be seen by a specialist statistical reviewer?

No

Is it clear how to make all supporting data available?

Not Applicable

Is the supplementary material necessary; and if so is it adequate and clear?

Not Applicable

Do you have any ethical concerns with this paper?

No

Comments to the Author

This has dramatically improved. I have just few comments concerning the text in the attached file

Decision letter (RSOB-19-0010.R0)

18-Jan-2019

Dear Dr Ikeda

We are pleased to inform you that your manuscript RSOB-19-0010 entitled "Functional ectodermal organ regeneration as the next generation of organ replacement therapy" has been accepted by the Editor for publication in Open Biology. The reviewer(s) have recommended publication, but also suggest some minor revisions to your manuscript. Therefore, we invite you to respond to the reviewer(s)' comments and revise your manuscript.

Please submit the revised version of your manuscript within 14 days. If you do not think you will be able to meet this date please let us know immediately and we can extend this deadline for you.

- 1) A text file of the manuscript (doc, txt, rtf or tex), including the references, tables (including captions) and figure captions. Please remove any tracked changes from the text before submission. PDF files are not an accepted format for the "Main Document".
- 2) A separate electronic file of each figure (tiff, EPS or print-quality PDF preferred). The format should be produced directly from original creation package, or original software format. Please note that PowerPoint files are not accepted.
- 3) Electronic supplementary material: this should be contained in a separate file from the main text and meet our ESM criteria (see <http://royalsocietypublishing.org/instructions-authors#question5>). All supplementary materials accompanying an accepted article will be treated as in their final form. They will be published alongside the paper on the journal website and posted on the online figshare repository. Files on figshare will be made available approximately one week before the accompanying article so that the supplementary material can be attributed a unique DOI.

Online supplementary material will also carry the title and description provided during submission, so please ensure these are accurate and informative. Note that the Royal Society will not edit or typeset supplementary material and it will be hosted as provided. Please ensure that the supplementary material includes the paper details (authors, title, journal name, article DOI). Your article DOI will be 10.1098/rsob.2016[last 4 digits of e.g. 10.1098/rsob.20160049].

- 4) A media summary: a short non-technical summary (up to 100 words) of the key findings/importance of your manuscript. Please try to write in simple English, avoid jargon, explain the importance of the topic, outline the main implications and describe why this topic is newsworthy.

Images

Data-Sharing

It is a condition of publication that data supporting your paper are made available. Data should be made available either in the electronic supplementary material or through an appropriate repository. Details of how to access data should be included in your paper. Please see <http://royalsocietypublishing.org/site/authors/policy.xhtml#question6> for more details.

Sincerely,

The Open Biology Team
<mailto:openbiology@royalsociety.org>

Referee:

Comments to the Author(s)

This has dramatically improved. I have just few comments concerning the text in the attached file

Author's Response to Decision Letter for (RSOB-19-0010.R0)

See Appendix A.

Decision letter (RSOB-19-0010.R1)

15-Feb-2019

Dear Dr Ikeda,

We are pleased to inform you that your manuscript entitled "Functional ectodermal organ regeneration as the next generation of organ replacement therapy" has been accepted by the Editor for publication in Open Biology.

Sincerely,

The Open Biology Team
mailto:openbiology@royalsociety.org

Appendix A

Referee's Comments to Author(s):

This has dramatically improved. I have just few comments concerning the text in the attached file.

This manuscript has greatly improved and for most part reads well. The only critique concerns the Abstract, and part 6 as well as the organoid issue. The difference between organoids, miniorgans, and functional organs should be clarified in the text (and tell that the organoids are usually composed of one tissue type, i.e. epithelium, and because stromal tissue is lacking, organoids cannot form functional organs). This issue should be covered already in the Introduction - it is mentioned there but should be expanded by moving the organoid issue from Part 6 (shortened) to Introduction. Also the Abstract should be rephrased: It gives

the impression that organoids could be used for bioengineering functional organs. lines 5-8 should be either removed or shortened.

Part 6 is partially quite unclear, and most of the content can be shortened and moved to Introduction. Part 6 could focus only on the cellular source of the stem cells for bioengineering organs: How to replace the organ-specific mouse embryonic cells? Here describe authors' "salivary gland from ES cells" experiments.

page 6, the title 2) Lachrymal- should read: lacrimal

Our Response: We have studied your comments carefully and found that you understood the value and significance of our study in this field. We are grateful for your evaluation and valuable suggestions for our manuscript. The potential application of regenerative medicine in the clinical setting is supported by the recent development of organ regeneration technique that mimic embryonic patterning and positioning signals, and thereby recapitulate the conditions required to facilitate organ self-organization during embryonic organogenesis. Follow the instructions of the referee, we have clarified the difference between organoids as miniorgans and

functional organs. We described that recent studies have reported the induction of an organ-forming-field types, and the subsequent generation of a range of organoids (i.e. 'mini-organs') of limited size, that partially recapitulate the organ function in vivo upon transplantation. Furthermore, we described about the studies that demonstrated the successful regeneration of a salivary gland, which were shown to be fully functional upon transplantation in a murine model. We have shortened Part6 and moves Introduction.

We have change Lachrymal to lacrimal on page 6.